# Evaluation of commercially available glucagon receptor antibodies and glucagon receptor expression

Anna Billeschou Bomholt [1], Christian Dall Johansen [1,2], Jens Bager Christensen [3], Sasha Alexandra Sampson Kjeldsen[1,2], Katrine Douglas Galsgaard[1,4], Marie Winther-Sørensen [1,2], Reza Serizawa[5], Mads Hornum[6,7], Esteban Porrini[8], Jens Pedersen [7,9], Cathrine Ørskov[1], Lise Lotte Gluud[7,10], Charlotte Mehlin Sørensen[1], Jens Juul Holst [1,4], Reidar Albrechtsen[11] & Nicolai Jacob Wewer Albrechtsen [1,2,12✉]

Glucagon is a major regulator of metabolism and drugs targeting the glucagon receptor (GCGR) are being developed. Insight into tissue and cell-specific expression of the GCGR is important to understand the biology of glucagon and to differentiate between direct and indirect actions of glucagon. However, it has been challenging to localize the GCGR in tissue due to low expression levels and lack of specific methods. Immunohistochemistry has frequently been used for GCGR localization, but antibodies targeting G-protein-coupled-receptors may be inaccurate. We evaluated all currently commercially available GCGR antibodies. The antibody, ab75240 (Antibody no. 11) was found to perform best among the twelve antibodies tested and using this antibody we found expression of the GCGR in the kidney, liver, preadipocytes, pancreas, and heart. Three antibody-independent approaches all confirmed the presence of the GCGR within the pancreas, liver and the kidneys. GCGR expression should be evaluated by both antibody and antibody-independent approaches.

[1] Department of Biomedical Sciences, Faculty of Health and Medical Sciences, University of Copenhagen, Copenhagen, Denmark. [2] Novo Nordisk Foundation Center for Protein Research, Faculty of Health and Medical Sciences, University of Copenhagen, Copenhagen, Denmark. [3] Department of Neuroscience, Faculty of Health and Medical Sciences, University of Copenhagen, Copenhagen, Denmark. [4] Novo Nordisk Foundation Center for Basic Metabolic Research, Faculty of Health and Medical Sciences, University of Copenhagen, Copenhagen, Denmark. [5] Department of Pathology, Copenhagen University Hospital, Hvidovre, Denmark. [6] Department of Nephrology, Centre for Cancer and Organ Diseases, Rigshospitalet, University of Copenhagen, Copenhagen, Denmark. [7] Department of Clinical Medicine, Faculty of Health Sciences, University of Copenhagen, Copenhagen, Denmark. [8] Instituto de Tecnologías Biomédicas, University of La Laguna, Research Unit, Hospital Universitario de Canarias, Tenerife, Spain. [9] Department of Internal Medicine, Endocrinology, Copenhagen University Hospital Herlev, Herlev, Denmark. [10] Gastro Unit, Copenhagen University Hospital, Hvidovre, Denmark. [11] Biotech Research & Innovation Centre, University of Copenhagen, Copenhagen, Denmark. [12] Department of Clinical Biochemistry, Bispebjerg and Frederiksberg Hospital, Copenhagen, Denmark. ✉email: nicolai.albrechtsen@sund.ku.dk

Glucagon is a peptide hormone secreted from pancreatic alpha cells and is a key regulator of numerous metabolic processes including glucose, protein, and lipid metabolism. These actions are mediated by the glucagon receptor (GCGR), a G-protein-coupled receptor (GPCR). Knowledge of the GCGR localization and thereby the sites of direct glucagon action is critical for the understanding of glucagon's physiology as well as its potential effects as a therapeutic agent. Besides in the liver, GCGR mRNA has been identified in the pancreas[1], central nervous system[2], heart tissue[1], adrenal glands[3], white and brown adipose tissue (BAT)[4,5], kidney[1,3], muscle[6], and spleen[1]. With Northern blot analysis in mice the presence of GCGR mRNA was narrowed down to the liver and kidney only, whereas GCGR mRNA by RT-PCR analysis was detected also in the small intestine, lung, brain, and pancreas[7]. In a rat study, GCGR mRNA was detected only in the liver and kidney, but not in adipose tissue, lung, heart, brain, or muscle[8]. Levels of GCGR mRNA and protein at the cell surface may not correlate since mRNA only represents the transcription and does not guarantee the presence of the mature protein[4]. Therefore, detection of GCGR protein and mapping of the GCGR protein distribution is essential. However, expression of GCGR protein is generally low and detailed mapping of GCGR protein distribution may therefore require sensitive antibody-based approaches including immunohistochemistry (IHC). Antibodies against GPCRs are potentially unreliable as reported for the glucagon-like peptide-1 receptor (GLP-1R)[9]. To deal with this problem we systematically evaluated twelve commercially available GCGR antibodies using HEK293 cells transfected with mouse or human GCGR cDNA transcripts and also studied liver sections from $Gcgr^{-/-}$ and $Gcgr^{+/+}$ mice. Based on the evaluation of the twelve available GCGR antibodies, one selected antibody, ab75240 (antibody no. 11) was subsequently used to evaluate GCGR localization in mice and humans. Autoradiography, RNA-sequencing, and single cell RNA-sequencing (scRNA-sequencing) were used as antibody-independent approaches to support the findings obtained with IHC.

## Results

**Antibody staining of transfected HEK293 cells.** We initially evaluated whether twelve commercially available antibodies (Table S1) were able to bind to human or mouse GCGR, expressed in HEK293 cells. The two GCGR transcripts harbored both a C-terminal cMyc-tag allowing us to double stain cells positive for the GCGR (Fig. 1). We used permeabilized cells, where the cells have been "pricked" open with methanol, to allow both intracellular as well as membrane epitope/antibody binding (Fig. 1). All GCGR antibodies together with the cMyc-antibody gave positive staining in 85% of the permeabilized HEK293 cells transfected with either the mouse or human GCGR transcript (Table S2). No staining was seen in the controls without primary antibodies (Fig. 1). The antibodies were also checked for membrane bound staining in HEK293 cells transfected with human GCGR. In this experiment cells were fixed with 4% paraformaldehyde instead of methanol. Ten out of the twelve GCGR antibodies positively stained non-permeabilized HEK293 cells transfected with the human GCGR, while antibodies no. 8 and 12 were negative (Fig. S1). The GCGR antibodies showed varying efficiency with respect to staining intensity.

**Evaluation of selected antibodies using liver tissue from Gcgr$^{+/+}$ and Gcgr$^{-/-}$ mice.** The twelve commercially available GCGR antibodies were then evaluated using formalin-fixed and paraffin-embedded liver sections from female $Gcgr^{+/+}$ and $Gcgr^{-/-}$ mice. Only antibody no. 11 revealed positive IHC staining of liver tissue from the $Gcgr^{+/+}$ mice (Fig. 2a). None of the remaining antibodies showed any specific staining of liver tissue from the $Gcgr^{+/+}$ mice. Antibodies no. 6, 7 and 11 showed unspecific binding in liver tissue from $Gcgr^{-/-}$ mice with varying intensity (Fig. 2a).

**Evaluation of selected antibodies using human liver tissue.** To ensure the applicability of these antibodies for human use, we also evaluated the performance of the twelve commercially available GCGR antibodies on paraffin-embedded liver tissue from patients with NASH. Antibodies no. 1, 2, 5, 8, and 9 did not bind to the human liver tissue, whereas antibodies no. 3, 4, 6, 7, 10, 11, and 12 showed varying staining intensity (Fig. S2).

Based on the antibody evaluations described above (the antibodies ability to generate a positive membrane staining of transfected HEK293 cells, to stain human and mouse liver tissue) we selected three GCGR antibodies (antibodies no. 4, 10, and 11) for further evaluation by western blotting.

**Evaluation of selected antibodies using Western Blotting.** The specificity of antibodies no. 4, 10, and 11 was further evaluated by performed western blotting of extracts of HEK293 cells transiently transfected with either human or mouse GCGR cDNA (Fig. 2b).

Bands corresponding to the approximately predicted size of GCGR (55 kDa) were observed using antibodies no. 10 and 11. Larger bands were detected at high intensity using antibody no. 10 whereas antibody no. 11 appeared more specific. Antibody no. 4 detected multiple bands and was not further evaluated (Fig. 2b). Mock-transfected HEK293 cells were used as negative control (Fig. S3a) and GAPDH was used as loading control (Fig. S3b). Antibody no. 11 was selected as the most specific.

To further validate the specificity of antibody no. 11, western blotting was performed on liver tissue from $Gcgr^{+/+}$ and $Gcgr^{-/-}$ mice (Fig. S4). Western blotting of liver tissue from $Gcgr^{+/+}$ mice confirmed the specificity of antibody no. 11, detecting a 55 kDa band, corresponding to the antibody/GCGR complex but also other bands were found and may reflect fragments due to proteolytic cleavage of the GCGR. Western blotting revealed no band from $Gcgr^{-/-}$ mice liver tissue (Fig. S4). An overview of the evaluation of the commercially GCGR antibodies are shown in Table S4.

**GCGR expression in various mouse tissue using immunohistochemistry.** Based on the results obtained from the validation of the twelve commercially available antibodies. Antibody no. 11 was selected for specific localization of GCGR expression in various tissue. Mouse tissue from the following organs were prepared for extensive GCGR immunostaining: liver, pancreas, kidney, brown and white adipose tissue (WAT), gastric mucosa, duodenum, ileum, colon, muscle, heart and adrenal gland (Fig. 3a).

The most intense immunostaining reactions were found in kidney tubuli (distal tubules) liver tissue, the islets of Langerhans in the pancreas, and heart muscle fibers. Glandular cells from stomach and enterocytes in crypts from the ileum were moderately stained and, in some areas, epithelial endocrine-like cells were stained. Duodenal as well as colonic epithelia were generally negative for GCGR or weakly positive. Muscle tissue was negative for GCGR. Concerning adipose tissue, the WAT was not stained, whereas preadipocytes and the BAT stained positive (Fig. S5). In addition, the cortical part of the adrenal gland, but not the medulla was stained for GCGR (Fig. 3a).

**GCGR expression in human kidney tissue using immunohistochemistry.** Based on the intense immunostaining of kidney tubular cells, human renal tissue from a healthy individual was also examined for localization of GCGR expression using antibody

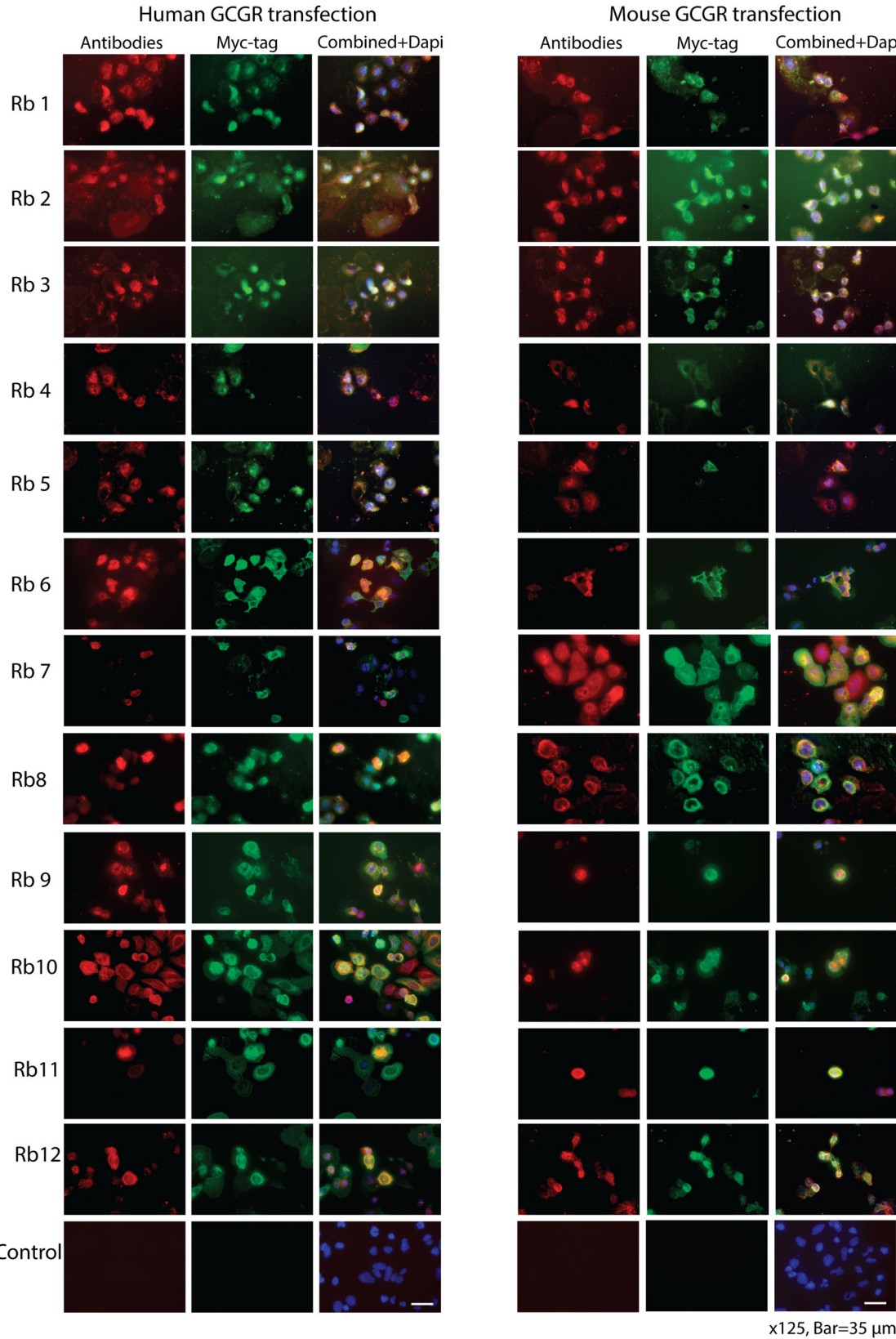

x125, Bar=35 µm

no. 11. Strong immunoreactivity was observed in the distal tubules. Cells within the proximal tubule were not stained (Fig. 3b).

**Co-staining of pancreatic tissue using immunohistochemistry.** GCGR expression in the pancreas is widely debated and we

therefore evaluated the cellular location of the GCGR, using antibody no. 11 and co-staining for either glucagon, insulin, or somatostatin. The co-staining with glucagon and antibody no. 11 suggested presence of GCGR in the alpha-cells. Co-staining with insulin and antibody no. 11 also revealed GCGR presence on beta-cells (Fig. 3c). Further, GCGR expression in the delta-cells

**Fig. 1 Antibody staining of transfected HEK293 cells.** Intra- and extracellular GCGR antibody binding of the twelve antibodies showed varying efficiency regarding staining of the cell and the cell membrane surface of methanol fixed, permeabilized HEK293 cells transiently transfected with human and mouse GCGR cDNA transcripts. Primary antibodies were used denoted as Rbx (x is the antibody number listed in Table S1). Control is transfected cells without primary antibody. Red color indicates GCGR antibody binding, green color is the Myc-tag on the GCGR and Dapi (blue/purple) indicates nuclei. X115, scale bar = 35 μm. Human GCGR vector: pCMV6-Entry (Cat# PS100001). Mouse GCGR vector: Cat# PS100001. See Table S2 for distribution of GCGR/c-Myc tag positive cells, and cells that were only GCGR or c-Myc tag positive. See also Fig. S1 for extracellular GCGR antibody binding of non-permeabilized HEK293 cells. Sequencing of vectors used for transient transfection are shown in Table S3. The 12 antibodies were diluted 1:100.

was also suggested by co-staining for somatostatin and antibody no. 11.

**Comparison of glucagon receptor expression by antibody and antibody-independent approaches**. Finally, we compared the findings obtained using the GCGR antibody no. 11 to results obtained using an antibody-independent approach: autoradiography. We choose autoradiography as it is a highly sensitive method that depends on ligand-receptor binding, in this case a $^{125}$I-labelled glucagon molecule. To control for unspecific binding we added, in a series of parallel experiments, a 1,000-fold excess of non-labelled glucagon to the $^{125}$I-Glucagon tracer thereby ensuring imbalanced competition to the GCGR of the non-labelled glucagon molecule allowing us to discriminate between true GCGR binding and unspecific binding.

High density of $^{125}$I-labeled glucagon grains was observed in hepatocytes, as expected, whereas in mice receiving both $^{125}$I-labeled glucagon and an excess of non-radioactive glucagon, a grain density corresponding to background density was observed (Fig. 4) supporting the specificity of the result. Similarly, high grain density was observed in the distal tubular and collecting duct cells of the kidney (Fig. 4), which was abolished in mice receiving excess non-labelled glucagon. These findings complement the positive staining of distal duct cells using antibody no. 11. When compared to both the surrounding exocrine tissue and to the mice who received an excess amount of non-radioactive glucagon, there appeared to be a weak increase in the number of grains in the islets of Langerhans.

The positive immunostainings of the gastric mucosa, heart, and adrenal glands all suggested GCGR receptor presence within these tissues, however, using autoradiography no grains accumulation was present within these tissues. This could indicate lack of receptor expression on the cell surface (Fig. 4) and/or intracellular accumulation of non-functional GCGR protein. However, an inferior sensitivity of the autoradiography approach cannot be excluded. Regarding tissue from the small intestine, muscles, adipose tissue, and spleen neither positive immunostaining nor autoradiografic grains were found (Fig. 4).

**RNA expression of the human GCGR from various tissues and specific cells**. To further supplement our findings by antibody and antibody-independent approaches we used RNA-sequencing data of adult human tissues that were made available by the GTEx consortium[10]. We selected tissues with at least ten available profiles sampled from donors who died from natural or violent causes and donors who died unexpectedly with a terminal phase of 1–24 h. This resulted in a total of 22 tissues and 5877 transcriptomic profiles (Table S5). Varying expression levels of GCGR mRNA were observed with the highest expression (transcripts per million (TPM)) in liver, kidney, and nerve tissue. GCGR mRNA expression in all other analyzed tissue was minimal or not present (Fig. 5a).

To increase our resolution and enable detection of cell specific GCGR mRNA expression, we used 10X Chromium scRNA-sequencing data from MacParland et al., 2018[11], Liao et al.,

2020[12] and Enge et al., 2017[13] for evaluating single-cell mRNA expression in liver, kidney and pancreas, respectively. These tissues were selected based on the bulk RNA-sequencing data and prior knowledge. The two data sets contain 8444, 23368 and 2466 cells, respectively, which passed the quality control restrictions. The analysis and cell type annotations of each cluster are based on the aforementioned publications. The distribution of cell types can be seen in Table S6. For liver tissue, GCGR expression was primarily observed in hepatocytes, whereas GCGR expression was mostly observed in beta-cells in the tissue of pancreatic islet (Fig. 5b). Generally, the GCGR expression of the kidney tissue is lower compared to the liver and pancreatic tissue, but the positive cells were primarily observed in collecting and distal tubule cells (Fig. 5b–d). Considering scRNA-sequencing being a tertiary approach to support the already confirmed GCGR expression in kidney via IHC, autoradiography, and bulk RNA-sequencing, the 10X data suggest that the expression originates from the collecting and distal tubule cells.

A summary on the evidence of GCGR expression in various tissue based on the reported data in this study and others are shown in Table 1.

## Discussion

Detection of GCGR expression using IHC, has been hindered by the lack of specific antibodies. In this study, we demonstrate that only one out of twelve commercially available antibodies was useful for staining of the GCGR, and based on antibody-independent approaches, we were able to confirm the findings of this antibody. Our study highlights the difficulty regarding IHC specificity in attempts to localize GPCRs[14–16].

The antibodies used in this study are all polyclonal. It is assumed that polyclonal antibodies are more likely to cross-react with other proteins than monoclonal antibodies. However, polyclonal antibodies have for diagnostic applications, been demonstrated to be advantageous due to their multi-epitope binding properties[17,18], which has been shown to increase sensitivity when used for detecting proteins that are present in low quantities[19]. Besides specificity, other factors have been demonstrated to influence the binding of antibodies to their targets, such as methods of cell/tissue fixation, tissue processing, and detection methods[20]. Especially fixation has a major impact on affinity and selectivity of antibodies since fixation changes the chemical property of the tissue, which alters the three-dimensional protein conformation by cross-linking[21].

In an attempt to localize the GCGR, many studies have resorted to methods such as RT-PCR[3,6], ribonuclease protection assay[1], or northern blotting analysis[7] for estimation of GCGR mRNA levels. The expression of GCGR mRNA was reported in various tissues. It has, however, been suggested that the GCGR mRNA and protein expression at the cell surface do not necessarily correlate, since the major regulation of the latter in vivo takes place at a posttranscriptional level[8].

The present findings suggest that additional, non-antibody-based validation approaches should be applied together with the antibody-based approaches to make up for possible issues with cross-reactivity and selectivity.

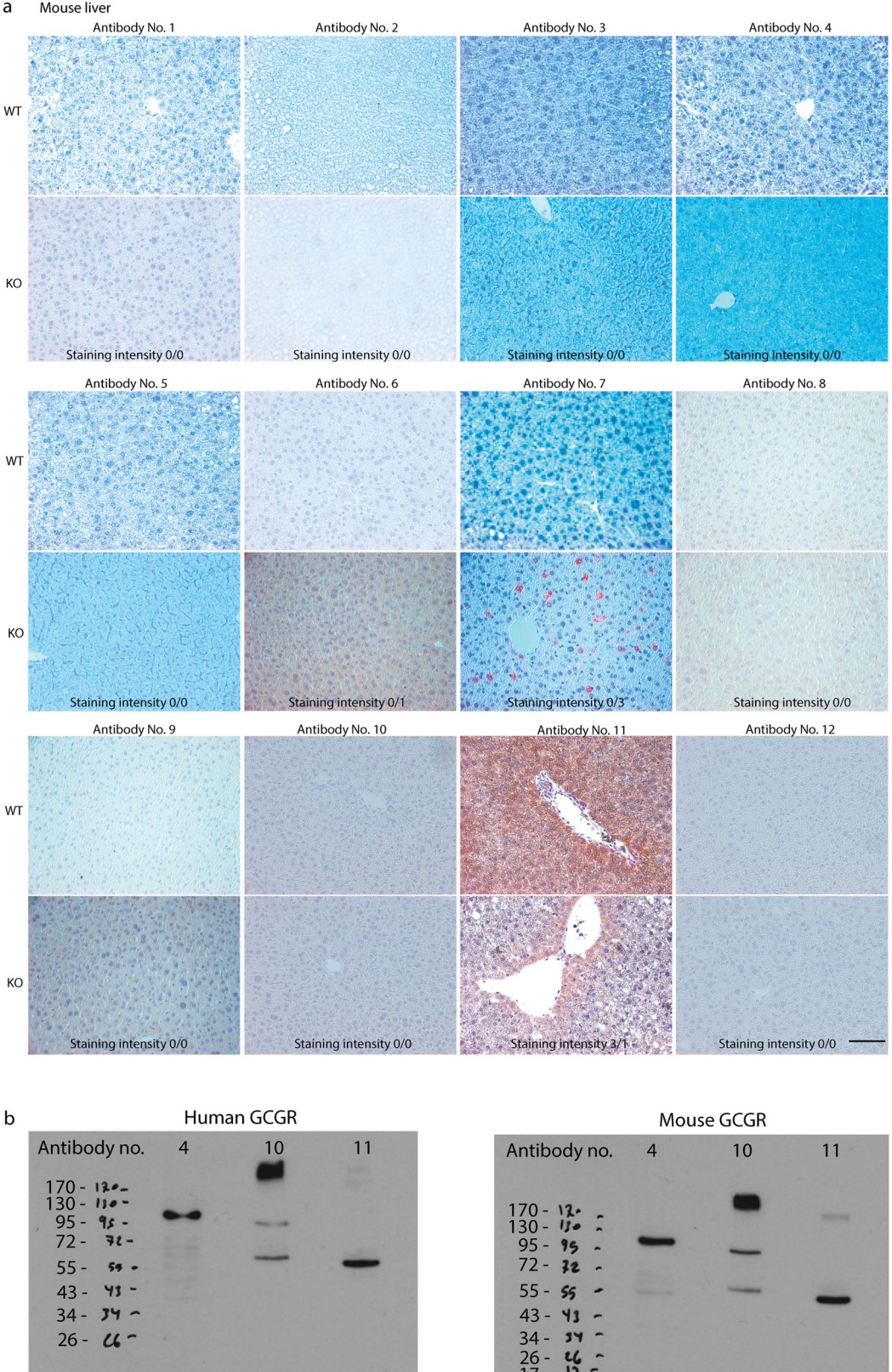

**Fig. 2 Evaluation of selected antibodies. a** Immunohistochemical staining of liver tissue from either the glucagon receptor wildtype ($Gcgr^{+/+}$) or glucagon receptor knockout ($Gcgr^{-/-}$) female mice, 8 weeks of age, using the 12 GCGR antibodies (1:100). Staining intensity scores between 0 and 3 of liver tissues from both the $Gcgr^{+/+}$ and $Gcgr^{-/-}$ mice are presented, the higher the score, the more receptor-antibody binding. X40, scale bar = 50 μm. The observed differences in background in the slides are suspected to be related to hematoxylin staining. **b** Uncropped western blots of transfected cells with either human glucagon receptor (GCGR) cDNA or mouse GCGR cDNA. Western blotting performed with selected antibodies no. 4, 10 and 11 (1:500). Uncropped western blots of a mock transfection showing a negative staining with antibody no. 11. and a western blot with loading control GAPDH are shown in Fig. S3.

Since glucagon primarily acts on liver GCGR to increase blood glucose by promoting glycogenolysis and gluconeogenesis[22], most studies focus on the GCGR function in the liver. However, GCGR has also been reported to be expressed in many other tissues, including the kidney, heart and adipose tissue[4,23]. In this study, we determined the specific location of the GCGR using an

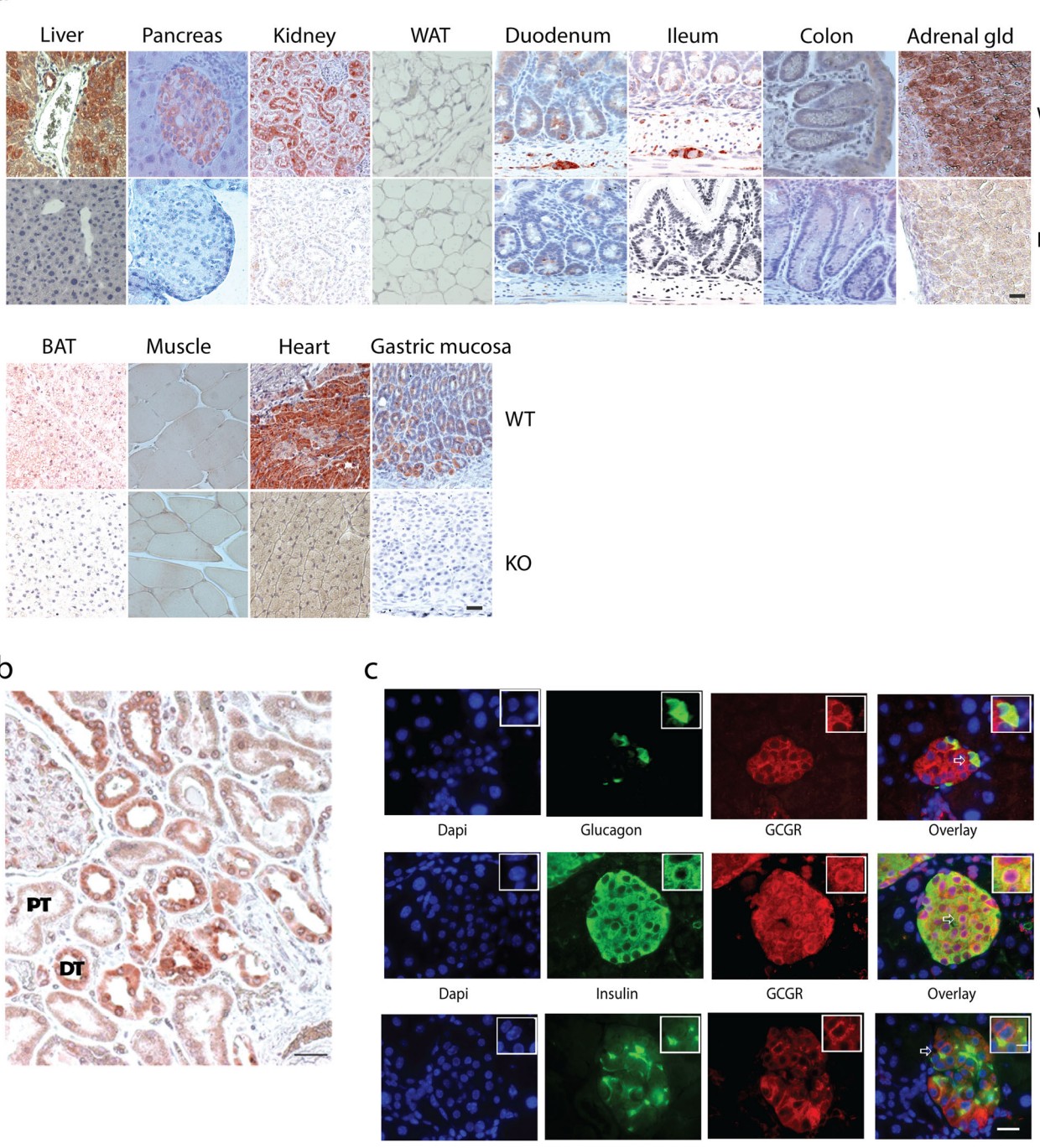

**Fig. 3 GCGR expression using immunohistochemistry. a** Immunohistochemical staining of various tissues from glucagon receptor wildtype (*Gcgr*$^{+/+}$) and knockout (*Gcgr*$^{-/-}$) female mice, 8 weeks of age, using the selected antibody no. 11 (1:100). Positive immunostainings were found in following tissues: heart, pancreas, kidney, liver, gastric mucosa, duodenum, ileum, brown adipose tissue (BAT), and the cortex of the adrenal gland, with a corresponding negative staining of *Gcgr*$^{-/-}$ tissue. No immunostaining was observed in the colon, white adipose tissue (WAT) and muscle tissue. Mouse liver, kidney, pancreas, WAT, colon, ileum, duodenum, adrenal gland, X100, scale bar = 30 μm. BAT, heart, muscle, gastric mucosa, X75, scale bar = 40 μm. See also Fig. S5 for immunohistochemical staining of premature adipocytes. **b** Antibody staining of healthy human renal tissue from autopsy using the selected antibody no. 11 (1:100). Cells within the proximal tubule (PT) were not stained, whereas cells in the distal tubules (DT) showed immunoreactivity toward the glucagon receptor (GCGR). X100, scale bar = 100 μm. **c** Co-staining of pancreatic mouse tissue, from *Gcgr*$^{+/+}$ female mice 8 weeks of age, with antibody no. 11 (1:100) and either glucagon (Scbt/Cat no. sc-514592), insulin (mybiosource/cat no. MBs448113), or somatostatin (Everestbiotech/cat no. EB11971). DAPI staining of nuclei. X200, scale bar = 20 μm. Bar insert = 6 μm, X400. Arrows indicate cells that were co-stained.

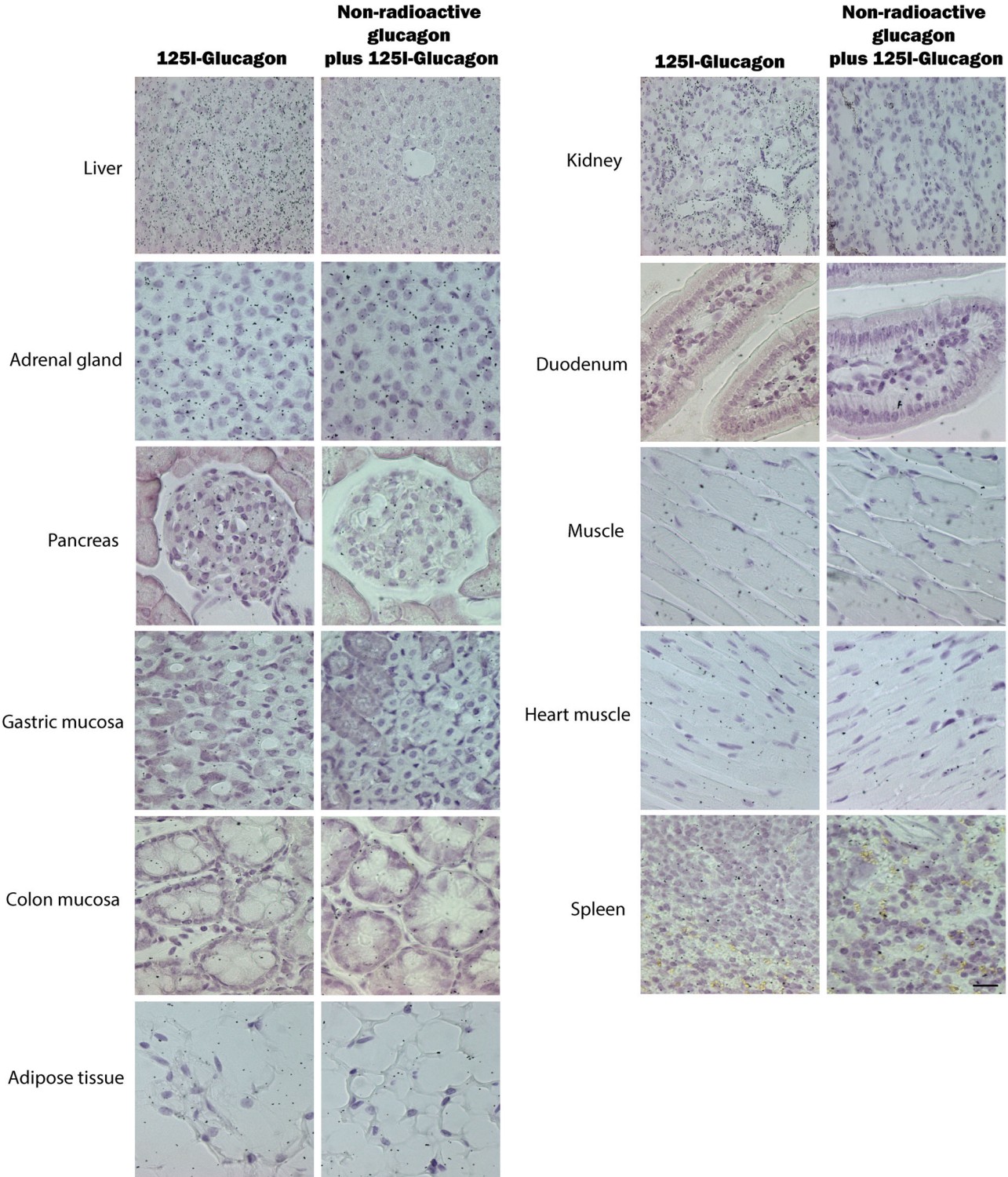

**Fig. 4 Comparison of glucagon receptor expression by antibody and antibody-independent approaches.** Microphotographs of the liver, kidney, adrenal gland, duodenum, pancreas, muscle tissue, gastric mucosa, heart, colon mucosa, spleen, and adipose tissue of female C57BL/6JRj mice, 13 weeks of age, that received an intravenous injection of 3 pmol $^{125}$I-Glucagon or 5 nmol of non-radioactive glucagon in combination with 3 pmol of $^{125}$I-Glucagon to test for specific binding. Autoradiography of mouse liver, kidney and pancreas showed high density of grains in the tissue, which was attenuated in mice receiving both labelled and non-labelled glucagon. Liver, adrenal gland, pancreas, gastric mucosa, colon mucosa, and fat tissue, X100, scale bar = 50 μm. Kidney, duodenum, muscle, heart muscle, and spleen, X70, scale bar = 70 μm.

antibody-based approach in combination with two non-antibody-based approaches: autoradiography and data analysis from RNA-sequencing data. Using IHC, various mouse tissues were examined for GCGR expression. The most intense immunostainings

were observed in the liver, heart muscle fibers, kidney tubuli (distal tubuli), the islets Langerhans in the pancreas, pre-adipocytes found in relation to kidney tissue and BAT. The specific GCGR localization in some of these different tissues has

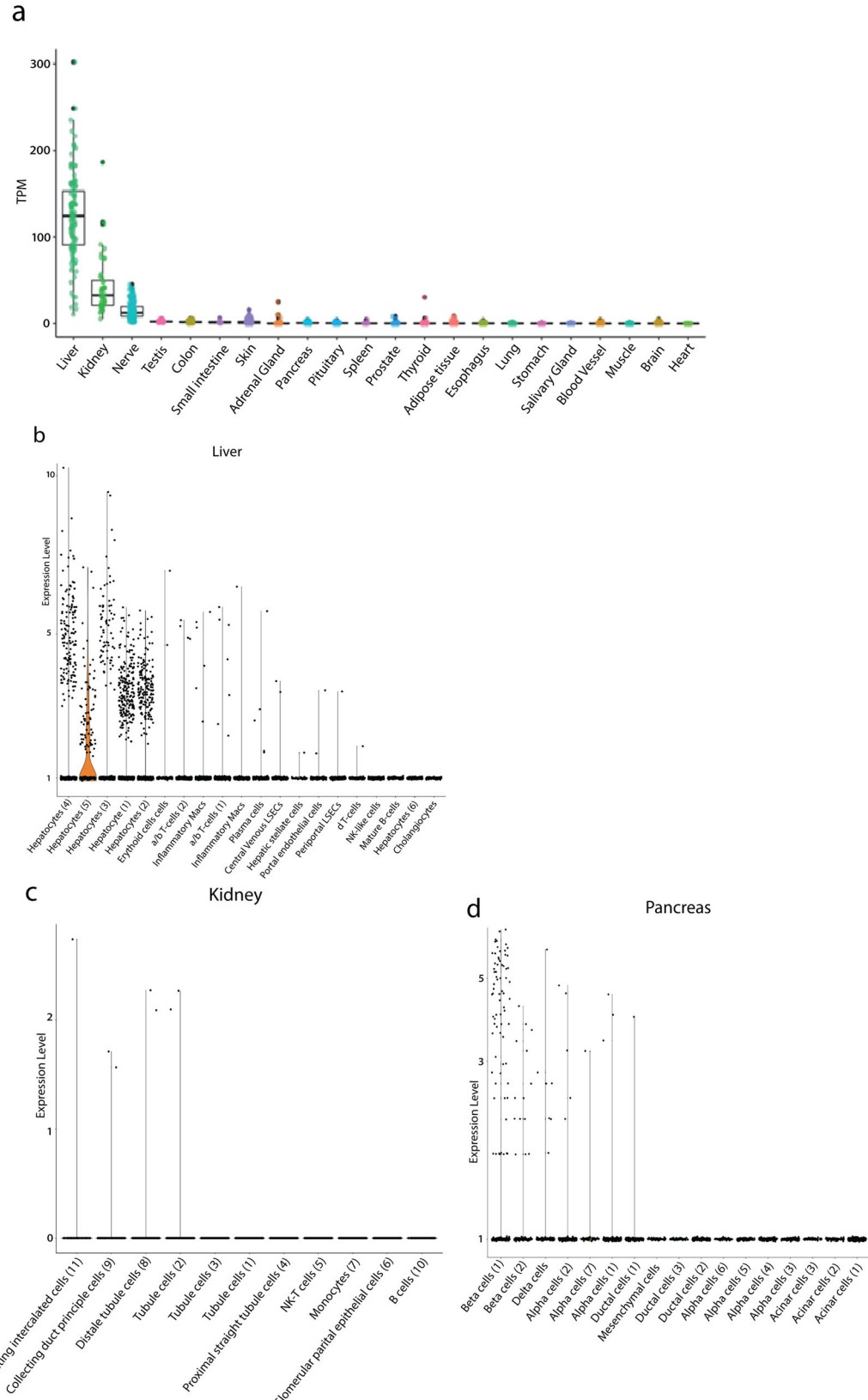

been the subject of vivid discussions. Profound disagreements regarding direct glucagon action and GCGR localization are apparent.

In the pancreas, the presence of the GCGR has previously been demonstrated, by measuring GCGR mRNA[1] and by sRNA-sequencing[24]. Controversies exist whether the insulinogenic effect

of glucagon is mediated directly via the GCGR or the GLP-1R on beta-cells. By IHC the GCGR, apart from the beta-cells, has also been identified in rodent pancreatic alpha- and delta-cells[25], suggesting that these cells may be influenced by glucagon in a paracrine manner and that glucagon might even bind to receptors on the alpha-cell, in an autocrine manner[26]. This agrees with the

**Fig. 5 RNA expression of the human GCGR from various tissues and specific cells. a** RNA-sequencing data generated by the Genotype-Tissue Expression (GTEx) project[10] from human tissues displayed in transcript per million (TPM) values for the glucagon receptor (GCGR). GCGR distribution across the dataset is visualized with boxplots, shown as median, 25th, and 75th percentiles. Points are displayed as outliers if they are above or below 1.5 times the interquartile range. TPM values of the individual samples are presented with the boxplots. The tissues are arranged according to average expression levels from left to right. See Table S5 for number of samples per tissue used. **b** Single-cell RNA-sequencing data of human liver, **c** kidney tissue and **d** islet of pancreatic tissue generated by MacParland et al., 2018[11], Enge et al., 2017[13] and Liao et al., 2020[12], respectively. The normalized expression of GCGR was evaluated in individual cells from the three data sets. The trend of GCGR expression was visualized in violin plots arranged by the average expression in each cell type in decreasing order.

results found in this study where antibody no. 11 demonstrated the presence of the GCGR in the alpha- and beta- cells, using co-staining for glucagon or insulin.

There are ongoing controversies about GCGR expression in adipose tissue[27]. Localization of the GCGR has been reported for both white and BAT[5,28], predominantly based on the identification of GCGR mRNA transcripts[1,4]. These studies generally observed a greater abundance of GCGR mRNA in brown compared to WAT. It has been suggested that glucagon increases blood flow, lipolysis, glucose uptake and oxygen consumption in white and BAT[29–31]. Studies in mouse and humans have not revealed a lipolytic effect of glucagon[27,32] whereas this may occur in rats[33]. In the present study we found GCGR staining of pre-adipocytes and BAT, while no WAT was stained, using antibody no. 11. This difference could be due to different methodologies where the PCR detects the GCGR mRNA transcripts. Similar to our findings, another study observed negative immunoblots of adipose tissues, when using a monoclonal GCGR antibody[34]. These observations could in fact be due to low expression levels of protein GCGR. However, we were also unable to identify mRNA GCGR transcript from the RNA sequencing data of adult human tissues processed within this study, which did not reveal expression of GCGR mRNA in adipose tissue. These findings are in alignment with previous finding, where GCGR was found in a range of human adipose tissue cell types using Single cell nucleus RNA-sequencing data[35]. Together, our results and those of others suggest that GCGR is not present in WAT of mice.

The adipose tissue is not the only questionable GCGR localization. We located the GCGR in the heart muscle fibers, using an IHC approach, but we were unable to detect the GCGR using the antibody-independent approach, autoradiography. Furthermore, it was not possible to identify GCGR mRNA transcripts from the RNA-sequencing data of adult human heart tissue. In previous studies, GCGR mRNA transcripts have been detected in heart tissue of both rats[4] and mice[36], however, not in humans[37]. GCGR mRNA has been detected from both right and left atria and ventricle of the adult mouse heart, using RT-PCR[36], where others have found GCGR distribution considerably higher in ventricular than in atrial myocardium by using western blotting[38]. Therefore, the understanding of glucagon action and specific GCGR localization in the heart remains limited.

In the kidney tissue, we observe specific GCGR localization in the distal tubule in human tissue, using antibody no 11. The presence of GCGR in renal tissue was supported by autoradiography, where [125]I-labeled glucagon grains were observed in distal tubular and collecting duct cells of the kidney. RNA-sequencing data confirmed GCGR presence in liver and kidney tissue while the scRNA-sequencing data clearly indicates that the GCGR is expressed in most types of hepatocytes, while the much lower expression of GCGR in kidney seems to be restricted to cells of collecting ducts and distal tubules. The ambiguity of the kidney scRNA-sequencing data might be due to either limited coverage, insufficient sequencing depth, or simply very low GCGR expression. The expression of GCGR in the distal part of the kidney supports a physiological role for glucagon in this organ.

In summary, GCGR localization remains challenging due to the lack of suitable commercially available GCGR antibodies. However, based on extensive validation of the IHC approach and the use of supplementary approaches, the localization of the GCGR could be estimated. Given sequence differences of the mouse and human GCGR our validation of GCGR antibodies in mouse models may not be translatable to human tissue although we have in this study included human liver tissue to circumvent this potential issue.

The present study identifies GCGR localization sites in the kidney, liver, pancreas, preadipocytes and heart muscle fibers. Further studies are however warranted to investigate GCGR expression in other tissues including the brain.

## Methods

**Lead contact**. Further information and requests for resources and reagents should be directed to the Lead Contact, Nicolai J. Wewer Albrechtsen (nicolai.albrechtsen@sund.ku.dk).

**Animal studies**. Female C57BL/6JRj mice were obtained from Janvier Laboratories (Saint-Berthevin Cedex, France). Glucagon receptor knockout (Gcgr[−/−]) mice C57BL/6J[Gcgrtm1Mjch] were described previously[39]. Homozygotes (Gcgr[−/−]) and wild-type (Gcgr[+/+]) littermates were used and both were breed with permission from Dr. Maureen Charron and obtained from the local breeding facility. Mice were housed in groups of maximum eight in individually ventilated cages and followed a light cycle of 12 h (lights on 6 AM–6 PM) with ad libitum access to standard chow (catalogue no. 1319, Altromin Spezialfutter, Lage, Germany) and water. Mice were allowed a minimum of one week of acclimatization before being included in any experiment. Female Gcgr[−/−] and Gcgr[+/+] mice (8 weeks of age) and female C57BL/6JRj mice (13 weeks of age) where used. Animal studies were conducted with permission from the Danish Animal Experiments Inspectorate, Ministry of Environment and Food of Denmark, permit 2018-15-0201-01397, and in accordance with the European Union Directive 2010/63/EU and guidelines of Danish legislation governing animal experimentation (1987), and the National Institutes of Health (publication no. 85–23). All studies were approved by the local ethical committee

**Human liver and kidney samples**. Human liver tissue was obtained with approval from the Danish Ethics Committee (H-18052725).

Human kidney tissue was obtained from the European Nephrectomy Biobank (ENBiBA) project (centres from Tenerife and Denmark). Danish Ethics Committee (H-19062684) and Institutional Review Board (IRB) of the Hospital Universitario de Canarias, Spain, approved the project.

**Antibodies and reagents**. Twelve GCGR antibodies were obtained from a number of different companies (Table S1). For final experiments the tested antibodies were diluted 1:100 for IHC/IF and for western blot 1:500. The vendors claimed the antibodies to react with GCGR in either human tissue, mouse tissue, or both (Table S1 for specifications). All antibodies were polyclonal and purified from rabbit immune sera. The secondary antibodies used were horseradish peroxidase (HRP)-conjugated, goat anti-rabbit, and rabbit anti-goat immunoglobulins from DAKO A/S (Glostrup, Denmark). Myc-DYKDDDDK Tag monoclonal antibody was obtained from Invitrogen (Naerum, Denmark). Alexa Fluor 546-conjugated goat anti-rabbit IgG (Fab)2 fragment from Thermo Fisher Scientific (Roskilde, Denmark) and Alexa Fluor 488 conjugated goat anti-mouse IgG (cat# no. A-11029) from Thermo Fisher Scientific. All primary antibodies were diluted in PBS with 5% normal goat serum. The secondary antibodies were diluted 1:1000 in PBS. All other chemicals were from Merck KGaA (Darmstadt, Germany).

**Preparation of human and mouse GCGR transfected HEK293 cells**. Human embryonic kidney (HEK) 293 cells stably expressing the vitronectin receptor αVβ3 integrin, called 293-VnR, were previously described[40]. Full-length human

**Table 1 Overview of glucagon receptor expression by antibody and antibody-independent approaches.**

| | Presence of GCGR using IHC staining with antibody no. 11 | Presence of GCGR using a radiolabeled ligand | Presence of GCGR using RNA-sequencing | Studies located GCGR using immunostaining | Studies located GCGR using a radiolabeled ligand |
|---|---|---|---|---|---|
| Liver | Yes | Yes | Yes (scRNA-sequencing indicates specific location primarily within hepatocytes) | Previously located in rats, using monoclonal antibody[34] | Previously located in mice[51,52] |
| Kidney | Yes | Yes | Yes (scRNA-sequencing suggests specific location within collecting and distal tubule cells) | Previously located in rats, using monoclonal antibody[34] | Located in In vitro[53-55] |
| Pancreas | Yes | Yes | Yes (scRNA-sequencing indicates specific location primarily within Beta-cells) | Previously located in rats and INS1 cells[25] | Previously located in In rats and INS1 cells[25] |
| Duodenum | No | No | No | | |
| Ileum | Yes | - | No | | Previously located in mice, small intestine[51] and dog intestinal smooth muscle cells[56] |
| Heart | Yes | No | No | | Previously located in mice[51] |
| Fat Tissue | Yes (Preadipocytes and BAT) | No | No | Previously located in rats, using monoclonal antibody[34] | Previously located in mice[51] |
| Adrenal gland | Yes (cortical part) | No | No | | |
| Gastric mucosa | Yes | No | No | | Previously located in mice[51] |
| Colon | No | No | No | Previously located in rats and INS1 cells[25] | Previously located in rats and INS1 cells[25] |
| Nerve tissue | - | - | Yes | | |

An overview of the GCGR expression by antibody and antibody-independent approaches on various tissues. Results from the present study, as well as results from other studies are presented.

(RC211179) and mouse GCGR (MR207767) both Myc-DDK-tagged cDNA were obtained from OriGene Technologies, Inc. (Rockville, MD 20850,USA) and inserted into the pcDNA3.1 expression vector. Adherent 293-Vnr cells were transfected with either human or mouse GCGR, using X-tremeGENE9 Transfection Reagent (Roche Applied Science, Hvidovre, Denmark). The cells were cultured in a humidified atmosphere of 21% $O_2$ and 5% $CO_2$ at 37 °C. Blot analysis of total cell lysates was performed to investigate expression levels of the constructs. Mock transfected cells were used as negative controls. Plasmids were sequenced to ensure the right transcript was used (Table S3).

**Immunofluorescence staining of transfected cells**. Visualization of GCGR by immunofluorescence staining was performed as previously described[41]. In brief, the transfected HEK293 cells were cultured for 2 days, fixated and permeabilized by cold methanol for 10 min at 4 °C or for non-permeabilized cells with 4% paraformaldehyde. Fluorescence imaging was performed using an inverted Zeiss Axiovert 220 Apotome system. The images were processed using the Axiovision program (Carl Zeiss, Oberkochen, Germany) and MetaMorph software. For nuclei staining, 4′, 6-diamidino-2-phenylindole (DAPI) was used (Invitrogen; 1:5000). The glucagon receptor antibodies were diluted 1:100.

**Tissue preparation and immunohistochemistry**. For preparation of paraffin-embedded tissue samples for chromogenic IHC, tissue from two female $Gcgr^{-/-}$ and $Gcgr^{+/+}$ mice (8 weeks of age), liver biopsies from individuals with non-alcoholic steatohepatitis (NASH) and renal tissue biopsies from healthy individuals, were fixed in 10% neutral formalin buffer (BAF-5000-08A, Cell Path Ltd, Powys, United Kingdom) for at least 24 h at 4 °C. Tissue samples from the heart, pancreas, kidney, liver, gastric mucosa, duodenum, ileum, colon, epididymal adipose tissue, interscapular BAT and adrenal gland were included. Tissues were dehydrated and paraffin-embedded and histological sections of 3 μm were cut. Sections were deparaffinized in xylene and rehydrated using descending alcohol solutions. The sections were transferred in epitope retrieval buffer (Dako, Denmark) and treated in a microwave oven for 2 × 10 min with 20 min pause in between. Endogenous peroxidase was blocked using a hydrogen peroxidase block. Next, a blocking buffer, 5% goat serum in PBS, was added to improve sensitivity by reducing background interference. The sections were incubated with the antibodies overnight at 4 °C in a humidified chamber and biotinylated goat anti-rabbit was added as secondary antibody. To visualize the GCGR positive cells, horseradish peroxidase, alongside its 3,3′diaminobenzidine substrate (Vector Lab, Burlingame, Ca) was added to the sections. For nuclear staining, haematoxylin was used. The glucagon receptor antibodies were diluted 1:100.

**Co-staining of pancreatic tissue using immunohistochemistry**. The double immunofluorescence staining of normal mouse pancreas islets of Langehans was performed using rabbit anti-GCGR, antibody no. 11, diluted 1:100 in 10% (v/v) fetal bovine serum in PBS (Cat# ab75240, 330 Cambridge Science Park, Cambridge, CB4 0FL, UK) in combination with following antibodies, goat anti-Insulin, diluted 1:100 in 10% (v/v) fetal bovine serum in PBS (Cat# MBS448113. MyBioSource, Inc. P.O. Box 153308San Diego, CA 92195-3308, USA), or goat anti-somatostatin, diluted 1:100 in 10% (v/v) fetal bovine serum in PBS (Cat# EB11971, Nordic Biosite, Toldbodgade 18, 5. DK-1253 Copenhagen, Denmark), or mouse mono-clonal IgG₁ κ anti-glucagon, diluted 1:100 and conjugated to Alexa Fluor® 488, (Cat# sc-514592, Santa Cruz Biotechnology, Inc. Bergheimer Str. 89-269115 Heidelberg, Germany). The sections were incubated overnight at 4 °C, washed thoroughly with PBS, and then double labeled with donkey anti-rabbit IgG Alexa Fluor 546 conjugate, diluted 1:1000 (Cat# A10040, ThermoFischer Scientific, Slangerup, Denmark) and donkey anti-goat IgG diluted 1:1000 AlexaFluor 488 Sections conjugate (Cat# A11055, ThermoFischer Scientific) after 30 min. incubation at room temperature. Cell nuclei were stained with DAPI (4′,6-Diamidine-2′-phenylindole dihydrochloride). Fluorescence imaging was performed using a Zeiss confocal 510 microscope equipped with a water 63×/1.0 Plan-Apochromat objective and Zen software (Carl Zeiss, Oberkochen, Germany).

**Western blot analysis**. Total cellular protein extraction was performed by standard methods[42]. Briefly, tissue and or cell line extracted cells were lysed in radioimmunoprecipitation assay buffer containing 50 mM Tris/HCl and 140 mM NaCl and the pH adjusted to 7.5. The following were added to the buffer:1% Triton X-100, 0.5% SDS,1 mM EDTA,1 mM PMSF, 1 Tablet of Protease Inhibitor Cocktail (Complete, EDTA-free, Roche). The extract was clarified by centrifugation (14,000 $g$ for 30 min at 4 °C). Extracted protein concentration was measured using the Bradford Comassie blue method (Pierce chemical Corp, USA). Protein samples were mixed with reducing sodium dodecyl sulfate (SDS) buffer and separated by 7.5% SDS-polyacrylamide gel electrophoresis, followed by electro-blotting. Non-specific binding was blocked with a 5% nonfat milk solution. The membrane was incubated with primary antibody overnight at 4 °C followed by incubation with a horseradish peroxidase-conjugated goat antibody. Protein bands were visualized using enhanced chemiluminescence (Amersham Bio-sciences, Amersham, UK). Housekeeping gene GAPDH was used as internal control to ensure equal loading. The glucagon receptor antibodies were diluted 1:500.

**Autoradiography**. Non-fasted female C57BL/6JRj mice (13 weeks of age) were anaesthetized by intraperitoneal injection of ketamine/xylazine (0.1 ml/20 g; ketamine 90 mg/kg (Ketaminol Vet.; MSD Animal Health, Madison, NJ, USA); xylazine 10 mg/kg (Rompun Vet.; Bayer Animal Health, Leverkusen, Germany)). When the animals were sufficiently sedated (absence of reflexes), the vena cava was exposed with a midline incision, and the mice received an injection of 3 pmol $^{125}$I-Glucagon, dissolved in PBS, over a 15 s period (Novo Nordisk, Bagsvaerd, Denmark). Half of the mice also received a 1000-fold excess (5 nmol) of non-radioactive glucagon in combination with $^{125}$I-Glucagon (Bachem, Switzerland) in the same injection to test for specific binding. Before injection, 10 μl of the $^{125}$I-Glucagon stock solution was counted in a γ-counter to determine the amount of radioactivity injected into each animal. After 10 min, the mice were euthanized by cutting the diaphragm to induce pneumothorax and the vascular system was perfused through the left cardiac ventricle (outlet through the right atrium) with saline to ensure removal of blood from the organs. Immediately after, the mice were fixated by perfusion for 2 min with fixative. The small intestine, heart, spleen, liver, kidney and adrenal gland as well as fat and muscle tissue were removed and post fixed in the same solution for 24 h. Tissue samples were embedded in paraffin and sections of 4 μm were coated with 43–45 °C Kodak NTB emulsion (VWR, Herlev, Denmark) diluted 1:1 with 43–45 °C water, dried, and stored in light-proof boxes at 5 °C for 6–8 weeks. Sections were then developed using Kodak D-19 developer (VWR) for 5 min, dipped 10 times in 0.5% acetic acid, and fixed in 30% sodium thiosulfate for 10 min. Sections were dehydrated, stained with hematoxylin and examined with a light microscope. Images were taken with a camera (Zeiss Axioscope 2 plus, Brock & Michelsen, Birkerød, Denmark) connected to the light microscope.

**Tissue-level gene expression analysis in humans**. Gene-level transcript per million (TPM) values were obtained from the GTEx Portal (version 8) and were produced as described on the GTEx documentation page (https://commonfund.nih.gov/hubmap). In brief, RNA-sequencing was done using Illumina TruSeq library construction protocol (non-stranded, polyA+ selection). This consisted of quantification of total RNA using the Quant-iTTM RiboGreen®RNA Assay Kit, polyA selection using oligo dT beads, heat fragmentation, and cDNA synthesis. Following end-repair and poly(A)-tailing, adaptor ligation was performed with Broad Institute-designed indexed adapters. RNA-sequencing libraries were paired-end sequenced (2 × 76 bp) on either an Illumina HiSeq2000 or an Illumina HiSeq2500 instrument according to the manufacturer's protocols. Reads were aligned to the human genome (GRCh38/hg38) using STAR (version 2.5.3a), based on the GENCODE v26 annotation. TPM values were produced with RNA-SeQC (version 1.1.9) using the *strictMode* flag.

We considered only samples from donors who died fast from natural or violent causes and donors who died unexpectedly with a terminal phase of 1–24 h. The data were filtered for tissues with <10 samples and for some tissues with low expression (Cervix Uteri, Bladder, Breast, Ovary, Uterus, Vagina, and Blood), which resulted in 5877 transcriptomic profiles across 22 tissues (Table S5). Boxplots showing the distribution of TPM values for GCGR across tissues were created using ggplot2[43] (version 3.3.5) in R (version 4.1.0).

**Cell-level gene expression analysis**. The normalized hepatic single-cell RNA-sequencing data were obtained from the GitHub related to MacParland et al., 2018 (https://github.com/BaderLab/HumanLiver) by use of MacParland's shiny app. Normalization and clustering were performed by MacParland et al., 2018 using Seurat[44–47]. While MacParland et al., 2018[11] used t-SNE as dimension reduction for data visualization, the default Seurat UMAP reduction algorithm was used in this publication. An elbow plot was visually inspected to ensure enough dimensions were included in the UMAP reduction.

The raw renal scRNA-sequencing data was available from the GitHub related to Liao et al., 2020 publication (https://github.com/lessonskit/Single-cell-RNA-sequencing-of-human-kidney). The SeuratObject was created differently compared to the Liao et al., 2020 script, as the criteria for *min.cells* and *min.features* were decreased allowing for detection of the lowly expressed GCGR. Quality control, normalization, and clustering were performed as in the original paper. In addition, the R package Harmony (https://CRAN.R-project.org/package=harmony)[48] (version 0.1.0) was used to adjust for batch correction and meta-analysis, using settings provided in Liao et al., 2020 "methods" section.

The pancreatic scRNA-sequencing data from Enge et al., 2017[13] was obtained via Azimuth (https://commonfund.nih.gov/hubmap), a part of the NIH Human Biomolecular Atlas Project (https://commonfund.nih.gov/hubmap). Normalization, cell cycle regression, and clustering was performed according to the standard Seurat pipeline, using the optimal clustering resolution based on calculated silhouette score. The exact parameters used are available on the GitHub related to this publication. The clusters were annotated with their respective cell type based on expression of key markers disclosed in Enge et al. 2017[13] and the Azimuth database.

Featureplots, dimplots, and violin plots were performed using built in Seurat functions (FeaturePlot, DimPlot, and VlnPlot, respectively) with parameters available on the GitHub related to this publication. Table S6 was created using dplyr[49] (version 1.1.3) in R (version 4.1.0).

**Statistics and reproducibility**. To increase reproducibility, we have provided all key reagents in Table S6.

In general, we have repeated experiments three times (technical replicates). We have not used statistical testing in this study. Data processing is described above.

**Reporting summary**. Further information on research design is available in the Nature Portfolio Reporting Summary linked to this article.

## Data availability
High resolution images and raw files are available through https://doi.org/10.17894/ucph. bac4efc1-3ccb-46ad-bbb4-62f5e35fd516. All RNA-seq data are publicly available[11–13] and processing of the data are made available through https://doi.org/10.5281/zenodo.7108958. Uncropped western blots are shown in Fig. 2; Fig. S3 and Fig. S4. Additional data sets generated are not publicly available but can be made available, upon reasonable request, by the Lead Contact, Nicolai Wewer Albrechtsen (nicolai.albrechtsen@sund.ku.dk).

## Code availability
Script for data processing of single-cell and bulk RNA-sequencing is available at GitHub (https://github.com/nicwin98/GCGR_Expression) and at https://doi.org/10.5281/zenodo. 7108958 [50].

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

## Acknowledgements

A sincere thanks to Maureen J. Charron, Departments of Biochemistry, Obstetrics and Gynecology and Women's Health, and Medicine, Albert Einstein College of Medicine, New York for providing glucagon receptor knockout mice and their wild-type littermates. This study was supported by NNF Excellence Emerging Investigator Grant – Endocrinology and Metabolism (Application No. NNF19OC0055001), EFSD Future Leader Award (NNF21SA0072746) and DFF Sapere Aude (1052-00003B). Novo Nordisk Foundation Center for Protein Research is supported financially by the Novo Nordisk Foundation (Grant agreement NNF14CC0001). The authors thank the European Renal Association (ERA) for the European Nephrectomy biobank (ENBiBA).

## Author contributions

J.J.H., R.A., and N.J.W.A. conceived and planned the experiments. A.B.B., C.D.J., J.B.C., S.A.S.K., K.D.G., M.W.S., C.Ø., and R.A. carried out the experiments. R.A performed all stainings and western blots. C.D.J., J.B.C., and N.J.W.A. carried out data interpretation using database available RNA-sequencing and scRNA-sequencing data. A.B.B., S.A.S.K., K.D.G., M.W.S., J.P., C.Ø., C.M.S, R.A., and N.J.W.A. contributed to the interpretation of the results. M.H. and E.P., collected and provided healthy human kidney biopsies. R.S and L.L.G. performed and provided liver biopsies from patients with NASH. A.B.B., C.D.J., J.B.C., S.A.S.K., K.D.G., J.J.H., R.A., and N.J.W.A. drafted paper. N.J.W.A. is responsible for project administration. All authors provided critical feedback and helped shape the research, analysis and paper.

## Competing interests

The authors declare no competing interests.
