## [Peer Review File · Communications Biology]

Reviewers' comments:

Reviewer #1 (Remarks to the Author):

Whether GCGR is expressed in various tissues has been a topic of debate and therefore important to have a reliable antibody that can specifically detect GCGR protein. The study by Bomholt et al. systematically evaluated the currently available antibodies using immunostaining with GCGR-expressing HEK cells, *Gcgr*^{-/-} mouse tissues and human samples. The results were verified by autoradiography, RNAseq and scRNAseq. The manuscript presented a well-organised study following a logical order and it is great to learn that there is an antibody that can reliably test GCGR in biological samples.

I have no major comment on the manuscript. Minor comments are as follows:

1. Line 87-88 it should be noted that the data referred to are in Figure S1.
2. As a general comment throughout the manuscript, the resolution of all the figures is rather poor – I suggest the authors improve the figures.
3. It is unclear why the authors referred to GCGR molecular weight as 62kDa – it is widely accepted that it is 54kDa (this is also consistent with the western blot the authors showed).
4. Line 279-281 'It has been established that the amount of functional GCGR protein is lower than the GCGR mRNA ...' reference is needed to back up the statement.
5. It will be good if the authors can also briefly discuss the difference between human and rodent GCGR – this can be relevant for the correct use of antibodies.
6. It was indicated that Supplementary tables (2 and 3) were for figures 9 and 10 – which are not in the present manuscript.
7. It was mentioned 'STAR Methods', which is not present in the current manuscript.
8. Line 447. The O₂ concentration is very high – do you have any particular reason for this?

Reviewer #2 (Remarks to the Author):

This manuscript describes a well executed analysis of the efficacy of 12 different antibodies raised to Glucagon receptor (GCGR). GPCR receptors are notoriously difficult to characterise by immunolabelling and the authors have used a battery of different techniques to confirm their findings with the different antibodies. Antibody 11 (from AbCam) showed convincing immunolabelling on both tissue sections of mouse liver and cells transfected with the human or mouse receptor construct. Positive labelling in mouse liver, kidney, pancreatic islets, fat, muscle and adrenal gland and human kidney tubules was found. Notably, the glucagon receptor was identified in mouse alpha, beta and delta cells of islets. To confirm this localisation of the receptor autoradiography of mouse tissues exposed to I125 glucagon in vivo was performed. This tricky technique successfully identified glucagon/receptor complexes in liver and kidney tissues. However, less convincingly in other tissues. mRNA expression of the receptor in a human tissue bank was evident in hepatocytes but less significantly in most other tissues. It is a pity that "whole pancreas" was used rather than pancreatic islet digest where the mRNA is certainly expressed.

The discussion is overlong (could be reduced by 50%) and there is repetition of some statements from other parts of the manuscript and tables. However, this manuscript provides a timely and strong analysis of the glucagon receptor indicating and confirming putative roles of glucagon in tissues other than the liver and pancreatic islets.

Specific points

Figure 2 B The authors should consider using the uncropped western blots from the supplemental information in place of these much tidied images.

Figure 3 A sp- Ventricle

Figure 4. At this resolution it is not possible to see the silver autoradiographic grains. It might be

preferable to enlarge (and increase the resolution) of some of the images and put the remainder into the supplemental information. There are no scale bars on this image.

Figure 5. The information relating to single cell seq in B is almost unreadable. Since most of these cells types are not discussed elsewhere, these data could be lodged in supplemental. It's a pity that single cell seq of pancreatic islets was not included.

Table 1 is a good overview and does not need to be expanded at length in the text of the discussion. The lack of identification of mRNA in whole pancreas merely shows that mRNA cannot be detected in cells forming <10% of a tissue mass.

Reviewers' comments:

Reviewer #1 (Remarks to the Author):

Whether GCGR is expressed in various tissues has been a topic of debate and therefore important to have a reliable antibody that can specifically detect GCGR protein. The study by Bomholt et al. systematically evaluated the currently available antibodies using immunostaining with GCGR-expressing HEK cells, Gcgr^{-/-} mouse tissues and human samples. The results were verified by autoradiography, RNAseq and scRNAseq. The manuscript presented a well-organised study following a logical order and it is great to learn that there is an antibody that can reliably test GCGR in biological samples.

I have no major comment on the manuscript.

Author response: We thank the reviewer for hers/his time and for providing constructive and positive feedback.

Minor comments are as follows:

1. Line 87-88 it should be noted that the data referred to are in Figure S1.

Author response: Thank you for making us aware of this. We have now added in the manuscript that the data referred to are in Figure S1.

2. As a general comment throughout the manuscript, the resolution of all the figures is rather poor – I suggest the authors improve the figures.

Author response: We have now improved the quality (including magnification) of all figures.

3. It is unclear why the authors referred to GCGR molecular weight as 62kDa – it is widely accepted that is 54kDa (this is also consistent with the western blot the authors showed).

Author response: Thank you for making of aware of this mistake. We have now corrected this error throughout the manuscript.

4. Line 279-281 ' It has been established that the amount of functional GCGR protein is lower than the GCGR mRNA ...' reference is needed to back up the statement.

Author response: Sorry, this was inaccurate. We have removed the sentence mentioned. The study we later refer to regarding potential differences of mRNA and protein levels of GCGR is <https://pubmed.ncbi.nlm.nih.gov/7999047/>

5. It will be good if the authors can also briefly discuss the difference between human and rodent GCGR – this is can be relevant for the correct use of antibodies.

Author response: We have inserted a short paragraph in the discussion on this topic.

Below please find an alignment analysis (uniprot) between the human and mouse GCGR. The analyses shows that there are considerable differences. Furthermore, looking into PTMs

(glycosylation in particular) such variables may also affect the actual binding capacity of the GCGR antibodies.

CLUSTAL O(1.2.4) multiple sequence alignment

```

sp|P47871|GLR_HUMAN      MPPCQPQRP-LLLLLLLACQVPSAQVMDFLFEKWKLYGDQCHHNSLLPPPELVCN    59
sp|Q61606|GLR_MOUSE     MPLTQLHCPHLLLLLVLSCLPEAPSAQVMDFLFEKWKLYSDQCHHNSLLPPPELVCN    60
** * : * *****;* * :.*****;*****;*****;*****;*****

sp|P47871|GLR_HUMAN      RTFDKYSWPDTPANTTANISCPWYLPWHHKVQHRFVFKRCGPDGQWVRGPRGQPWRDAS  119
sp|Q61606|GLR_MOUSE     RTFDKYSWPDTPNNTANISCPWYLPWYHKVQHRLVFKRCGPDGQWVRGPRGQPWRNAS  120
*****;*****;*****;*****;*****;*****;*****;*****;*****

sp|P47871|GLR_HUMAN      QCQMDGEEIEVQKEVAKMYSSSQVMTYVGYSLSLGALLLAILGGLSKLHCTRNAIHAN  179
sp|Q61606|GLR_MOUSE     QCQLDDEEIEVQKGVAKMYSSSQVMTYVGYSLSLGALLLAILLGLRKLHCTRNYIHGN  180
**.*.***** ***** *****;*****;*****;*****;*****;*****

sp|P47871|GLR_HUMAN      LFAFVVKASSVLVIDGLLRTRYKIGDDLVSSTWLSGAVAGCRVAAVFMQYGIIVANY  239
sp|Q61606|GLR_MOUSE     LFAFVVKAGSVLVIDNLKTRYKIGDDLVSSTWLSGAVAGCRVAVFMQYGIIVANY  240
*****.* ***** **;*****;*****;*****;*****;*****;*****

sp|P47871|GLR_HUMAN      CWLLVEGLYLHNLLGLATLPERSSFSLYLGIGWGAPMLFVVPWAVVKCLFENVQCWTSND  299
sp|Q61606|GLR_MOUSE     CWLLVEGVYLYSLLSLATFSERSSFSLYLGIGWGAPLLFVIPWVVKCLFENVQCWTSND  300
*****;* * :.*****;*****;*****;*****;*****;*****;*****

sp|P47871|GLR_HUMAN      NMGFWILRFPVFLAILNFFIFVRIVQLLVAKLRARQMHHTDYKFRLLAKSTLTLIPLL G  359
sp|Q61606|GLR_MOUSE     NMGFWILRIPVFLALLINFFIFVHIIHLLVAKLRAHQMHYADYKFRLLARSTLTLIPLL G  360
*****;*****;*****;*****;*****;*****;*****;*****;*****

sp|P47871|GLR_HUMAN      VHEVVFVFTDEHAQGLRSTKLFDFLSSFGLLVAVLYCFLNKEVQAEIMRRWRQWQ  419
sp|Q61606|GLR_MOUSE     VHEVVFVFTDEHAQGLRSTKLFDFLSSFGLLVAVLYCFLNKEVQAEIMRRWRQWQ  420
*****;*****;*****;*****;*****;*****;*****;*****;*****

sp|P47871|GLR_HUMAN      LGKVLWEERNTSNHR---ASSSPGHGPPSKELQFGRGGGSQD---SSAETPLAGGLPRL  472
sp|Q61606|GLR_MOUSE     EGKALQEERLASSHGSHMAPAGPCHGDPCEKLQMSAGSSSGTGCVPSPMETSLASSLPRL  480
**.* *** ;.* * :.* * * * :.*** :.* * . * ** * * .****

sp|P47871|GLR_HUMAN      AESPF  477
sp|Q61606|GLR_MOUSE     ADSPT  485
**.*

```

6. It was indicated that Supplementary tables (2 and 3) were for figures 9 and 10 – which are not in the present manuscript.

Author response: Thank you for making of aware of this mistake. This has now been revised.

7. It was mentioned ‘STAR Methods’, which is not present in the current manuscript.

Author response: Revised

8. Line 447. The O2 concentration is very high – do you have any particular reason for this?

Author response: This was simply a typing error, it has been corrected to 21% O2.

Reviewer #2 (Remarks to the Author):

This manuscript describes a well executed analysis of the efficacy of 12 different antibodies raised to Glucagon receptor (GCGR). GPCR receptors are notoriously difficult to characterise by immunolabelling and the authors have used a battery of different techniques to confirm their findings with the different antibodies. Antibody 11 (from AbCam) showed convincing immunolabelling on both tissue sections of mouse liver and cells transfected with the human or mouse receptor construct. Positive labelling in mouse liver, kidney, pancreatic islets, fat, muscle and adrenal gland and human kidney tubules was found. Notably, the glucagon receptor was identified in mouse alpha, beta and delta cells of islets.

To confirm this localisation of the receptor autoradiography of mouse tissues exposed to I125 glucagon in vivo was performed. This tricky technique successfully identified glucagon/receptor complexes in liver and kidney tissues. However, less convincingly in other tissues. mRNA expression of the receptor in a human tissue bank was evident in hepatocytes but less significantly in most other tissues.

Author response: We thank the reviewer for hers/his time and for providing constructive and positive feedback.

It is a pity that “whole pancreas” was used rather than pancreatic islet digest where the mRNA is certainly expressed.

Author response: Yes, you are right. We have now performed analysis of single cell RNA sequencing data from pancreas and have included these results in the revised manuscript.

The discussion is overlong (could be reduced by 50%) and there is repetition of some statements from other parts of the manuscript and tables.

Author response: We have reduced the discussion considerably (from ~1800 words to ~1100 words).

However, this manuscript provides a timely and strong analysis of the glucagon receptor indicating and confirming putative roles of glucagon in tissues other than the liver and pancreatic islets.
Specific points

Figure 2 B The authors should consider using the uncropped western blots from the supplemental information in place of these much tidied images.

Author response: Thank you for this comment. The uncropped western blots originally from the supplemental information is now included in figure 2.

Figure 3 A sp- Ventricle

Author response: Revised

Figure 4. At this resolution it is not possible to see the silver autoradiographic grains. Its might be preferable to enlarge (and increase the resolution) of some of the images and put the remainder into the supplemental information. There are no scale bars on this image.

Author response: We have now improved the quality of the figures and we hope that it is now possible to see the autoradiographic grains on the images. Further, a scale bar has been added to the images.

Please also see our response to reviewer 1.

Figure 5. The information relating to single cell seq in B is almost unreadable. Since most of these cells types are not discussed elsewhere, these data could be lodged in supplemental. It's a pity that single cell seq of pancreatic islets was not included.

Author response: Yes, you are right. We have revised figure 5B by removing the dot plots and improving the quality of the images. Further, we have added scRNA results from pancreatic islet digest instead.

Table 1 is a good overview and does not need to be expanded at length in the text of the discussion. The lack of identification of mRNA in whole pancreas merely shows that mRNA cannot be detected in cells forming <10% of a tissue mass.

Author response: Thank you for this comment. We have added scRNA results from pancreatic islet digest instead showing GCGR expression in islets (primarily beta and delta cells).

Reviewers' comments:

Reviewer #1 (Remarks to the Author):

The authors have addressed most of my comments and the manuscript is much improved. It will be good if the authors can add one more series of panels in Figure 3c to show the double positive staining with more details (i.e., with the 'overlay' panels zoomed in at areas that are double positive).

Reviewer #2 (Remarks to the Author):

This is a much improved manuscript with adjustments made to all of the reviewers comments. The discussion is very much improved.

A few minor points:

Sp Summary

Insert a Glucagon is a major

Figure 4 caption:efficiency in methanol fixed,.....

The methods for the immunolabelling mentions secondaries to goat and rabbit but omits any details on the anti mouse secondary

Figure 2 a. The resolution is still not good on these images. You could consider rearranging them so that each image is a bit larger

Figure 4 caption Sp mouse liver.....

Reviewers' comments:

Reviewer #1 (Remarks to the Author):

The authors have addressed most of my comments and the manuscript is much improved. It will be good if the authors can add one more series of panels in Figure 3c to show the double positive staining with more details (i.e., with the 'overlay' panels zoomed in at areas that are double positive).

Author response: We thank the reviewer for hers/his time and for providing constructive and positive feedback.

Regarding figure 3c. We have in the revised manuscript added zoom panels for double positive staining. We hope this satisfy the reviewer.

Reviewer #2 (Remarks to the Author):

This is a much improved manuscript with adjustments made to all of the reviewers comments. The discussion is very much improved.

Author response: We thank the reviewer for hers/his time and for providing constructive and positive feedback.

A few minor points:

Sp Summary

Insert a Glucagon is a major

Author response: We have inserted "Glucagon is a major" in the first sentence in the summary. We hope we have understood the point by the reviewer correctly.

Figure 4 caption:efficiency in methanol fixed,.....

Author response: Thank you for making us aware of this mistake. Reviewer mention caption in figure 4, however, we believe that the reviewer might have meant figure 1. The sentence has now been revised.

The methods for the immunolabelling mentions secondaries to goat and rabbit but omits any details on the anti mouse secondary

Author response: Thank you for this comment. We have now added more details on the anti-mouse secondary antibody, concerning dilution, catalogue number and origin

Figure 2 a. The resolution is still not good on these images. You could consider rearranging them so that each image is a bit larger

Author response: Thank you for this comment. We have rearranged the images in figure 2a, so they now appear larger. We hope this satisfy the reviewer.

Figure 4 caption Sp mouse liver.....

Author response: We are not sure what the reviewer means by this comment but believe it deals with Figure 3 caption and have inserted 'mouse' in the sentence mentioning the liver. We hope this is what the reviewer meant.